# A computational approach to quantifying miscounting of radiation-induced double-strand break immunofluorescent foci

Samuel P. Ingram [1,2 ✉], John-William Warmenhoven[1,3], Nicholas T. Henthorn [1,3], Amy L. Chadiwck [1,3], Elham E. Santina[1,3], Stephen J. McMahon [4], Jan Schuemann [5], Norman F. Kirkby[1,3], Ranald I. Mackay[1,2], Karen J. Kirkby[1,3] & Michael J. Merchant[1,3]

Immunofluorescent tagging of DNA double-strand break (DSB) markers, such as $\gamma$-H2AX and other DSB repair proteins, are powerful tools in understanding biological consequences following irradiation. However, whilst the technique is widespread, there are many uncertainties related to its ability to resolve and reliably deduce the number of foci when counting using microscopy. We present a new tool for simulating radiation-induced foci in order to evaluate microscope performance within in silico immunofluorescent images. Simulations of the DSB distributions were generated using Monte Carlo track-structure simulation. For each DSB distribution, a corresponding DNA repair process was modelled and the un-repaired DSBs were recorded at several time points. Corresponding microscopy images for both a DSB and ($\gamma$-H2AX) fluorescent marker were generated and compared for different microscopes, radiation types and doses. Statistically significant differences in miscounting were found across most of the tested scenarios. These inconsistencies were propagated through to repair kinetics where there was a perceived change between radiation-types. These changes did not reflect the underlying repair rate and were caused by inconsistencies in foci counting. We conclude that these underlying uncertainties must be considered when analysing images of DNA damage markers to ensure differences observed are real and are not caused by non-systematic miscounting.

[1] Division of Cancer Sciences, Faculty of Biology, Medicine and Health, The University of Manchester, Oxford Rd, Manchester M13 9PL, UK. [2] Christie Medical Physics and Engineering, The Christie NHS Foundation Trust, Wilmslow Rd, Manchester M20 4BX, UK. [3] The Christie NHS Foundation Trust, Manchester Academic Health Science Centre, Wilmslow Rd, Manchester M20 4BX, UK. [4] Patrick G Johnston Centre for Cancer Research, Queens University Belfast, 97 Lisburn Rd, Belfast BT9 7AE, UK. [5] Massachusetts General Hospital and Harvard Medical School, Department of Radiation Oncology, 30 Fruit Street, Boston, MA 02114, USA. ✉email: samuel.ingram1@nhs.net

The manner in which radiation can damage living cells has resulted in the scientific interest in the driving mechanisms of radiation-induced cytotoxicity. In early experiments, interactions of radiation within the nucleus were identified as the primary drivers for radiation-induced cell death[1]. The DNA was found to be the sensitive target of the nucleus, this is predominately driven by the way radiation imparts energy within the molecular structure of the DNA causing structural damage. This DNA damage can manifest itself in various forms, such as base lesions, single- and double-strand breaks (DSBs) in the sugar-phosphate backbone[2]. The type of DNA damage that is formed has varying corresponding cellular lethality, with DNA DSBs being one of the most toxic[3]. Consequently, this has caused a large amount of interest in using DNA DSB induction as an indication of the cytotoxic characteristics of different radiation properties, such as dose, radiation particle and linear energy transfer (LET).

DNA damage response (DDR) signalling is a pivotal part of how adept cells are at preserving function following irradiation. It was discovered that the histone H2AX became phosphorylated ($\gamma$-H2AX) following the induction of DNA DSBs[4]. The $\gamma$-H2AX assay is a sensitive molecular marker for the building of an understanding of the initial DSB yields following irradiation and its subsequent DNA repair. This information has played a vital part in developing our current understanding of the mechanisms of DNA DSB formation[5] and how different radiation properties have different biological consequences[6–9]. Over time other key molecular targets, such as 53BP1, MDC1, ATM and proteins which form the MRN complex[10], have been sought out as alternative immunofluorescent markers of DNA DSB. Each molecular target has its role to play in the DDR, which can result in differences when inferring underlying mechanisms from them[11]. There has been an increased interest in using the DNA DSB markers in translational cancer research as a possible method of predicting patient-specific response and further optimising patient treatment[12–14]. However, whilst these approaches are enticing, with an interest in clinical application the limitations of the technique must be widely explored.

The developing understanding of the relationship between radiation and the cellular response has been driven by the applicability of radiotherapy in a large proportion of cancer treatment regimens. The efficacy of radiotherapy is dictated by the physical ability to sculpt the radiation dose to the tumour whilst sparing healthy tissue. The unavoidable radiation that is delivered to healthy tissue, specifically organs at risk (e.g. spinal cord), is what limits the maximum radiation dose that can be given to the tumour and therefore the tumour control probability from a treatment[15]. The quantitative measurement between the radiation that can be delivered to the tumour whilst being safe for the organs at risk is known as the therapeutic ratio. To utilise the experimental understanding of therapeutic ratio at a clinical level it is important to develop corresponding models which can further generalise predictions, to cover both the magnitude and spatial distributions of radiation doses delivered in treatment plans. However, these models are limited by the experimental uncertainties and the ability to generalise to populations of patients which are inherently biologically diverse[16]. Therefore, efforts have continuously been made in the development of broader and/or more detailed models of radiation response to overcome these limitations and better define the therapeutic ratio, ensuring treatment plans are optimal. There are emerging attempts to model radiation response mechanistically, this increases the complexity but aims to fully capture the mechanisms at play increasing the possibility of model generalisation. A subset of these developing models, include an explicit description of the induced DNA damage following irradiation[17–19] and some

include aspects of the DDR (e.g. DNA repair)[20–22]. These models enable us to simulate the spatial distribution of DNA DSBs using Monte Carlo frameworks such as Geant4-DNA[23] and TOPAS-nBio[24] for a range of incident radiation setups. The information provided by mechanistic models can be coupled with our understanding of immunofluorescence and microscopy to start to explore the limitations of the experimental technique[25,26].

In this study, we use radiation track-structure simulations[27] to produce representative spatial distributions of DNA DSBs and a bi-exponential repair model to predict the distribution at various time points. We then developed software called PyFoci to generate computational microscope images and look at the deviation in the foci detected and the known number of DSBs in the simulation. This software builds on previously developed models[25,26,28] through the addition of several mechanisms which improve its likeness to the experimental setting. PyFoci is able to both emulate the fluorescent foci tagging being either direct (Ku/DNA-PKcs markers) or indirect ($\gamma$-H2AX markers) to mark the spatial positions of the fluorescent antibodies. Indirect markers are controlled by explicit modelling of the chromatin conformation[29] and utilises recent insights of how H2AX becomes activated following DNA damage[30]. This is then convolved with the blurring caused by visualising sub-resolution fluorescent antibodies (~10 nm)[31] for specific models of a microscope at varying magnification and numerical aperture (NA). The emulated microscope images can be analysed using standard foci counting techniques, allowing for the quantification of miscounting between the actual number of DSBs present within the simulation and the number of DSB foci counted on the emulated microscope image. The amount of miscounting has been evaluated for a range of radiation types, including LETs and doses to highlight the difficulties of comparing experimental DSB fluorescence foci across different radiation properties. We show that the level of miscounting significantly changes in the majority of tested radiation types and doses, which subsequently leads to perceived changes in repair kinetics, even when no such differences are present in the simulations. Therefore, we propose PyFoci as a tool for experimentalists to enhance their studies by quantifying these often omitted uncertainties and reinforce caution when trying to make mechanistic conclusions based on fluorescent foci imaging. We also conclude that this approach should be used to adequately compare simulated levels of DNA DSB damage and the subsequent repair in mechanistic models with experimental data. The creation of a computational microscope image can be seen as an intermediate step to incorporate the deviation between absolute DNA DSBs and the number which would be represented via a fluorescent foci technique.

## Results

**Comparison of miscounting with Ku/DNA-PKcs markers.** Through the simulation of the microscope image, it becomes possible to evaluate DNA DSB miscounting. This is possible as there is a known amount of simulated DSBs within a single confocal slice of the cell nucleus and this can be compared to the number of identified foci. As foci are commonly used as surrogates for identifying a DSB one can calculate the miscount as the difference between foci detected and the actual number of DSBs. Positive values of miscounting would indicate over-counting, most likely due to the fluorescence of neighbouring slices being counted. Whereas, negative values of miscounting would indicate under-counting, which could be fluorescence from multiple DSBs not being distinguishable or the fluorescence being under the threshold value for the automated identification. The number of identified miscounts depends on the time point, radiation type, marker type and dose. To evaluate these relationships for the Ku/DNA-PKcs marker visualisation a series of box-plots (Fig. 1)

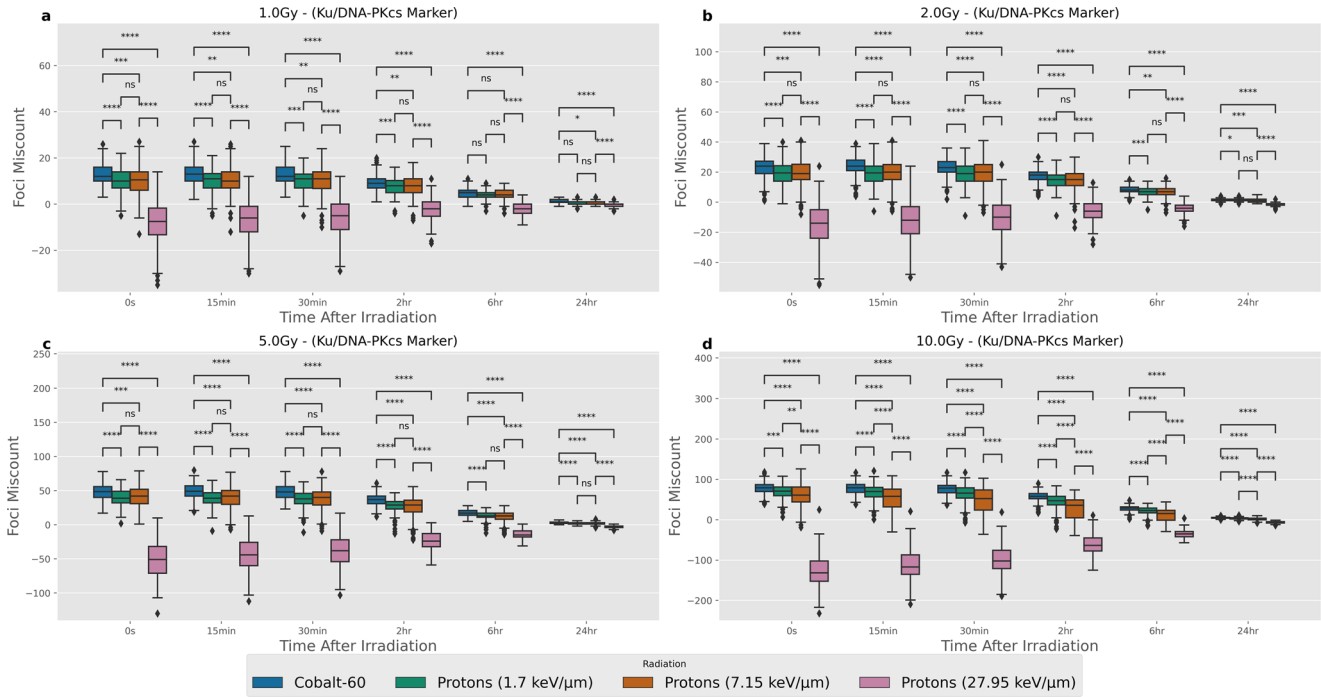

**Fig. 1 Distributions of miscounting between foci detected and the number of DSBs within the central microscope slice of the cell nucleus when using a Ku/DNA-PKcs marker of fluorescence.** Each panel is the same four radiation types and six-time points being compared for different radiation doses, where **a–d** correspond to 1.0, 2.0, 5.0, 10.0 Gy respectively. Error bars correspond to 1.5 times the interquartile range in either direction. Mann–Whitney test between different radiation types at each time point and dose to highlight statistically significant differences. $p$ values have been adjusted using the Bonferroni correction. Significance notation refers to the following thresholds: ns $= p > 0.05$, $*p > 0.01$, $**p > 1e{-}3$, $***p > 1e{-}4$, $****p < 1e{-}4$. There are 200 samples for each box-plot. All microscope images have been emulated using the Airyscan ×63 point spread function.

compare the amount of miscounting across different time points, radiation types and doses.

As it is common in the literature to evaluate the differences in fluorescent foci between radiation types a series of Mann–Whitney tests were performed to identify a statistically significant difference in miscounting at each time point and dose. All tests were significant in simulations at 10 Gy with the difference between protons at 1.7 keV/μm and 7.15 keV/μm being not significant between 1–5 Gy. There are also some non-significant differences seen between low LET protons (keV/μm) and Cobalt-60 for 1 Gy at late (>6 h) time points. The largest change between radiation types is with high-LET protons (27.95 keV/μm), there is a significant amount of under-counting, which increases with dose, that likely corresponds to DSBs becoming increasingly clustered and non-distinguishable. To get a perspective of the magnitude of these values the same graph has been presented as the percentage difference between foci detected and the number of actual DSBs in Fig. S3, where the same distinct under-counting is seen for high-LET protons. A feature of examining percentage foci miscounting is that at later time points there is a tendency to see an increased proportion of over-counting due to the contribution of neighbouring slices outweighing the rate of actual DSBs within the slice being repaired. In some experiments researchers may be interested in how counted foci change as a function of dose, where each dose used is compared, this was also evaluated for statistically significant changes in miscounting (Fig. S4). The majority of results demonstrated statistically significant differences when attempting to compare different doses of the same radiation type with increasing foci miscount for increased dose. However, when compared to the percentage difference for dose comparison (Fig. S5) it becomes apparent that whilst there is an increase in foci miscount for low LET radiations this does not translate to a distinct increase in percentage foci miscount.

**Comparison of miscounting with γ-H2AX markers.** The γ-H2AX marker is evaluated in the same manner as the Ku/DNA-PKcs marker for the same radiation types, time points and doses (Fig. 2). There is a consistently high number of statistically significant differences in miscounting over the tested parameters. However, the amount of under-counting in high-LET protons is reduced at later time points, with several median values indicating there is over-counting of foci when compared to the Ku/DNA-PKcs marker results. Whilst the amount of actual DSBs and the clustering is the same in both datasets the visualisation approach alters where the fluorescent light source is produced depending on the origin of the fluorescent signal. Within the γ-H2AX marker images, a single DSB results in multiple H2AX phosphorylations, each acting as a point of fluorescence, which in turn gives a larger area of fluorescence increasing the amount of contribution from breaks within neighbouring slices. Conversely, nearby DSBs within the same topologically associated domain (TAD) can provide activation to the same H2AX histones producing slightly larger foci, which are non-distinguishable. The amount of foci miscount tends to show over-counting more frequently suggesting the light contribution from neighbouring slices being detected as foci is greater than the DSBs within the confocal slice being under-counted. With high-LET, early time points centralising around zero foci miscount it is thought that at this point the over-counting of foci fluorescence from neighbouring slice is of a similar magnitude to the under-counting of foci in the confocal slice which is not being counted as individual foci. This is supported when examining the percentage foci miscount, which shows a similar centralisation around zero percentage foci miscount for high-LET early time points (Fig. S6). When making dose comparisons (Fig. S7) we see all scenarios which are significantly different. The corresponding analysis of the percentage foci miscount when comparing different doses suggest similar amounts of miscounting between doses (Fig. S8).

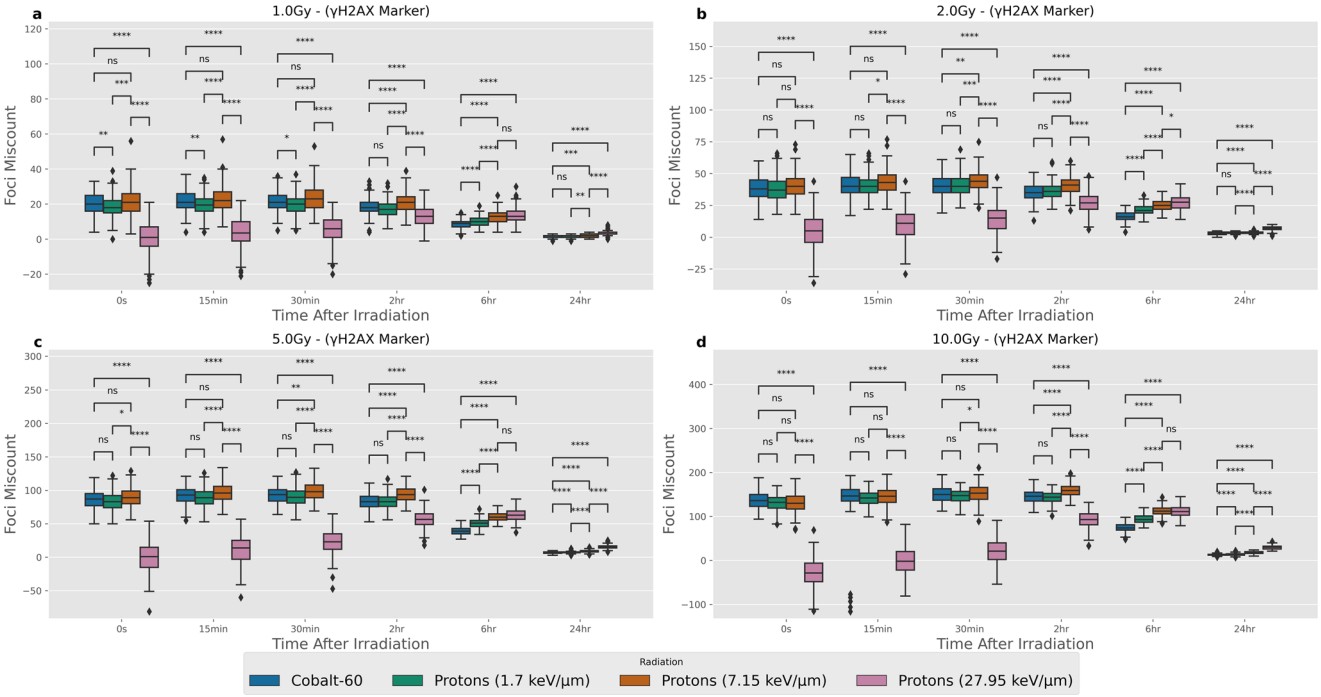

**Fig. 2 Distributions of miscounting between foci detected and the number of DSBs within the central microscope slice of the cell nucleus when using an γ-H2AX marker of fluorescence.** Each panel is the same four radiation types and six-time points being compared for different radiation doses, where **a–d** correspond to 1.0, 2.0, 5.0, 10.0 Gy respectively. Error bars correspond to 1.5 times the interquartile range in either direction. Mann–Whitney test between different radiation types at each time point and dose to highlight statistically significant differences. $p$ values have been adjusted using the Bonferroni correction. Significance notation refers to the following thresholds: ns $= p > 0.05$, $*p > 0.01$, $**p > 1e{-}3$, $***p > 1e{-}4$, $****p < 1e{-}4$. There are 200 samples for each box-plot. All microscope images have been emulated using the Airyscan ×63 point spread function.

**Implications for repair kinetics**. Typically, DSB fluorescence foci would be analysed as a function of time to examine the repair kinetics. When comparing across radiation types it is common to normalise to the first time point to account for the differences in the total number of DSBs. The normalised comparison of the radiation types for the actual number of DSBs and the identified foci in the Ku/DNA-PKcs marker and γ-H2AX marker images are presented in Fig. 3 for a dose of 2 Gy and using the Airyscan ×63 PSF. To evaluate the perceived effects of changes in repair kinetics that derive from miscounting alone the same repair rate was applied, in the form of the bi-exponential equation (Eq. (1)), to all radiation types. The differences within the DSB and γ-H2AX marker visualisations are given in Fig. 3b, c respectively. To better correspond to experimental set-ups the time normalisation point was chosen to be at 15 min as this would allow time for the DDR to recruit necessary proteins that would be observable in fluorescence experiments. The differences in the visualised repair kinetics are due to varying miscounting at subsequent time points, this results in the identical repair kinetics from Fig. 3a being observed as different when visualised via a DSB or γ-H2AX marker. The largest miscount for the Ku/DNA-PKcs marker (Fig. 1) and γ-H2AX marker (Fig. 2) was seen for high-LET proton and this is propagated to the largest observed effects. When using a Ku/DNA-PKcs marker (Fig. 3b) high-LET protons DSBs were increasingly under-counted (Fig. S3) which results in the perceived quickening of repair kinetics. Conversely, when using an γ-H2AX marker (Fig. 3c) DSBs are initially under-counted due to overlaps, but become over-counted due to signal from other layers at later time points (Fig. S6) resulting in the perceived slow down of repair kinetics. However, it should be acknowledged that increasingly research publications are avoiding normalisation as it is highly sensitive to deviations in the normalisation time point. The same plot without normalisation

can be seen in Fig. S9, whilst there are differences in the initial number of DSBs seen within the foci detection, these differences appear reduced when compared to the actual values. The overall, perceived faster and slower repair of high-LET protons is still visible for the DSB and γ-H2AX marker respectively.

**Effects of microscope magnification**. The emulation of the microscope visualisation is achieved by convolution with the experimentally derived PSF. This PSF is unique to each microscope and each magnification. The effects of the magnification on percentage foci miscount was evaluated for the Airyscan microscope for both the DSB and γ-H2AX marker visualisations at 2 Gy and 15 min post-irradiation time point (Fig. 4). To help give perspective for the percentage foci miscount the actual number of DSBs within the evaluated slice are provided within each box-plot as the white numeric value. The increased number of actual DSBs within a slice is a respective measure of how the larger z-slice resolution encompasses a larger volume of measurement. It is unlikely that different magnifications would be used in a single experiment, but in the event of comparing between experiments where different magnifications have been used, the quantification of miscounting may be of use. The percentage miscount is largely preserved across magnifications greater than ×10, with a significant under-counting at ×10. The Z-axis pixel spacing for ×10 is 1.14 μm (Table 1) which is a considerable proportion of the 2 μm thickness of the simulated flattened cell nucleus causing substantial spatial averaging increasing the likelihood of not distinguishing DSBs. Interestingly, the percentage foci miscount is similar between the other magnifications suggesting that normalised levels of foci may be comparable between different experiments where the same microscope, radiation and dose has been used, but different magnifications. To ensure this was not a feature of just the 15 min time point the same analysis was run on

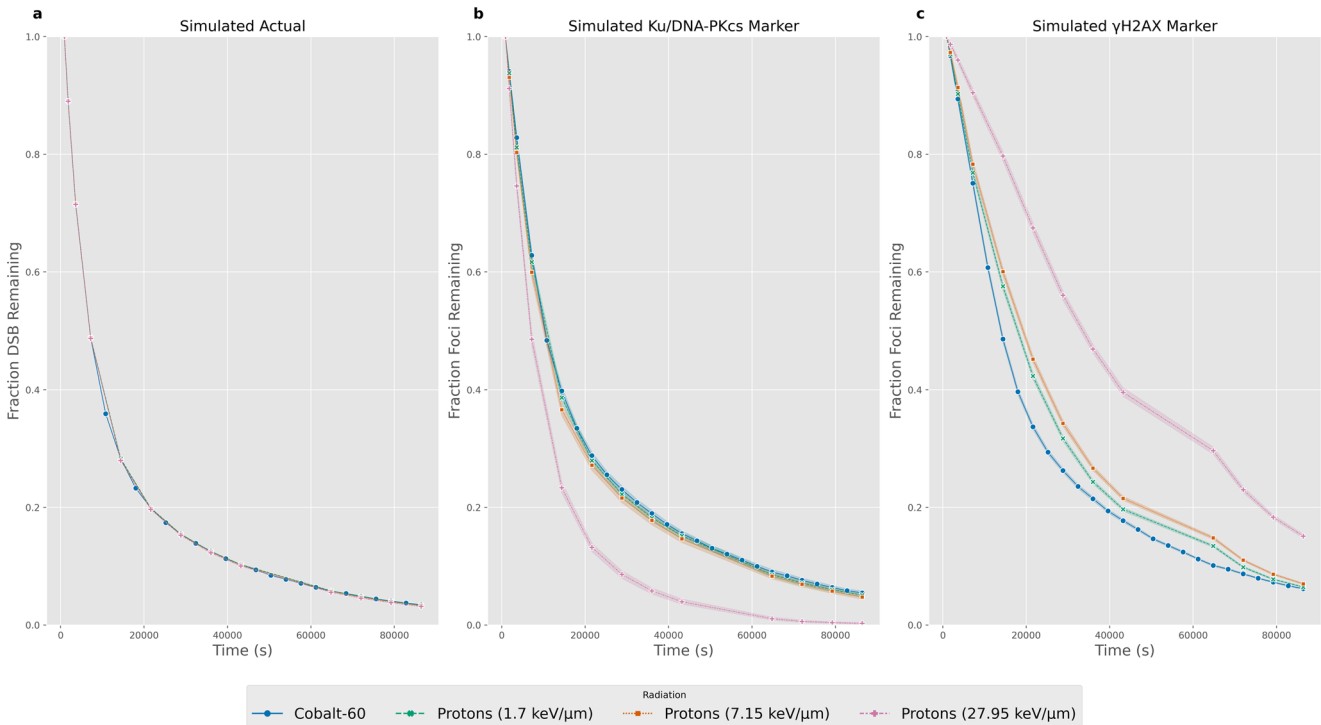

**Fig. 3 Repair kinetics comparison between virtual cells irradiated with four different incident radiation conditions.** The fraction of DSB/Foci remaining have been normalised to the 15 min time point to allow comparison of repair kinetics regardless of the total number of DSB breaks induced. **a** shows the repair kinetics of the actual number of DSBs within the evaluated microscope slice. The repair kinetics applied using the bi-exponential repair model is kept constant across all radiation types to separate out the impact of miscounting alone. **b** The repair kinetics when calculated from the foci detected in the Ku/DNA-PKcs marker microscope images. **c** The repair kinetics when calculated from the foci detected in the $\gamma$-H2AX marker microscope images. All microscope images have been emulated using the Airyscan ×63 and the results shown are for 2 Gy of radiation. Error bars are shown as banding and corresponds to the 95% confidence interval.

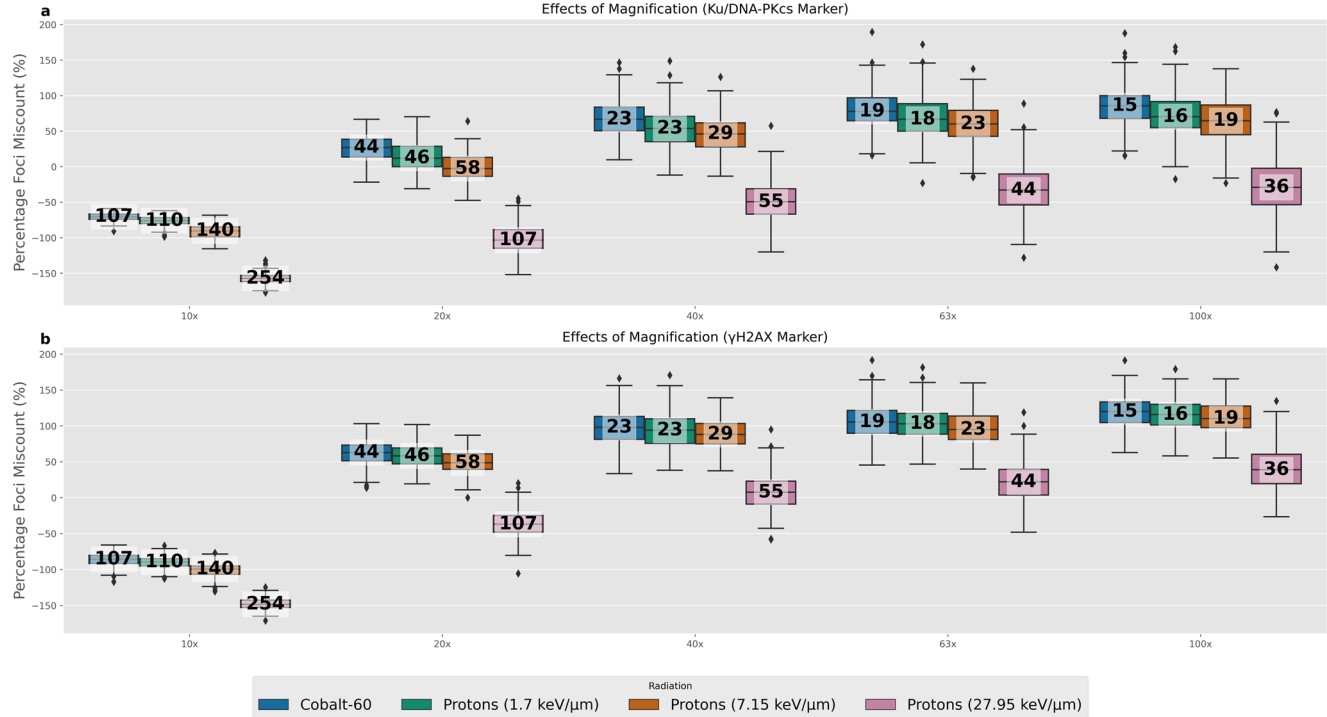

**Fig. 4 Comparison of percentage miscounting at different Airyscan magnifications.** The value shown in white within each box-plot is the actual number of DSBs within the evaluated microscope slice. This aids in giving perspective on the amount of miscounting each percentage corresponds to. **a** are the results when using a Ku/DNA-PKcs marker. **b** are the results when using an $\gamma$-H2AX marker. All microscope images are at 2 Gy dose and for the 15 min time point. There are 200 samples for each box-plot.

**Table 1 Microscope parameters—the pixel spacing for XY and Z (μm) that the resultant PyFoci microscope image uses.**

**Microscope parameters (XY pixel spacing, Z pixel spacing, numerical aperture)**

|  | ×10 | ×20 | ×40 | ×63 | ×100 |
|---|---|---|---|---|---|
| Airyscan | 0.13, 1.14, 0.45 | 0.059, 0.3, 0.8 | 0.035, 0.15, 1.3 | 0.033, 0.12, 1.4 | 0.028, 0.1, 1.46 |
| gSTED | – | 0.065, 0.3, 0.95[a] | 0.055, 0.20, 1.1 | 0.042, 0.13, 1.4 | 0.44, 0.13, 1.4 |
| Lowlight | – | 0.21, 0.5, 0.5 | 0.11, 0.3, 0.75 | – | 0.043, 0.1, 1.45 |
| MultiPhoton | 0.20, 1.0, 0.3 | 0.54, 0.25, 0.95[b] | 0.058, 0.2, 0.85 | 0.044, 0.15, 1.2 | – |
| Phenix | – | 0.3, 0.8, 1.0 | 0.15, 0.5, 1.1 | 0.095, 0.4, 1.15 | – |
| STED | – | 0.075, 0.4, 0.75 | 0.045, 0.15, 1.1 | 0.034, 0.1, 1.4 | 0.035, 0.1, 1.4 |

Along with the numerical aperture used when measuring the microscope PSF. Values given for all size microscopes at their respective magnifications. The STED microscope also has a PSF at ×25 which has the following values (0.06, 0.25, 0.95) for XY pixel spacing, Z pixel spacing and numerical aperture respectively.
[a]×23 magnification not ×20.
[b]×25 magnification not ×20.

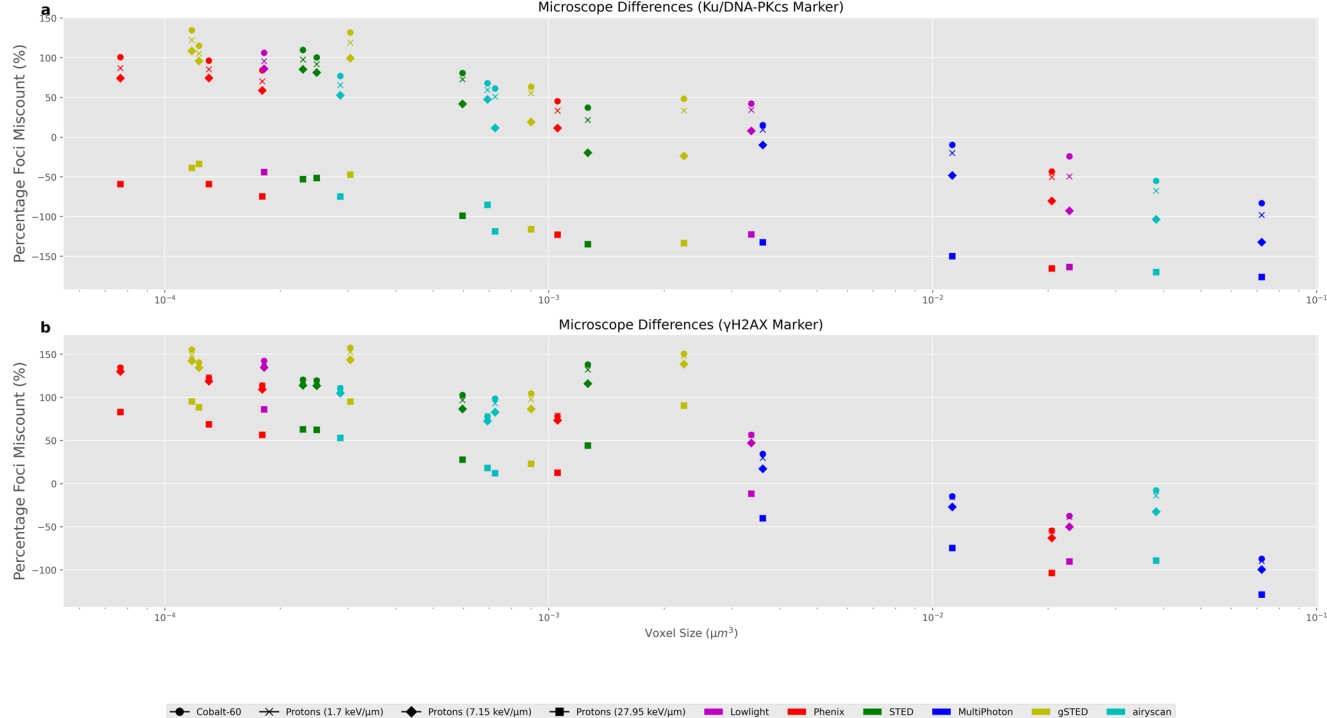

**Fig. 5 Comparison of percentage miscounting across all available microscope and magnifications.** Voxel size has been used as the descriptor of each microscope and is presented on a log axis. **a** are the results when using a Ku/DNA-PKcs marker. **b** are the results when using an γ-H2AX marker. Error bars are given as the standard error in the mean for the 1200 microscope images per point (200 cells and 6 time points). Error bars correspond to 1.5 times the interquartile range in either direction. All microscope images are at a 2 Gy dose.

the distribution of all time points combined (Fig. S10) where the overall trend persists, but the spread in the distribution increases due to the DSB repair giving a wider range of DSBs to being visualised for each box-plot.

**Effects of microscope resolution.** To evaluate the effects of different microscopes, including the various magnifications, the average percentage foci miscount was compared against voxel size across all 23 microscope/magnification configurations for 2 Gy dose (Fig. 5). The voxel size is calculated as the X*Y*Z pixel spacing which can be found in Table 1, small voxel sizes are typical for higher resolution microscopes at high magnification. There appears to be a negative relationship between voxel size and percentage foci miscount, with a decreasing value with increasing voxel size. The change in percentage foci miscounting indicates optimal foci counting relies on a balance between the tendency to over-count at small voxel sizes and under-count at larger voxel sizes. Over-counting is due to neighbouring slice

interference being larger at higher resolutions and the under-counting is the result of increased spatial averaging and a smaller influence of neighbouring slices.

**Impact of deconvolution on miscounting.** Emulating confocal microscopy allows for a single z-slice to be chosen whilst reducing out-of-plane fluorescence. However, as we have demonstrated, whilst the "intensity" may not be focused, with much of it being filtered out, there is still a contribution to the in-focus plane which can lead to foci over-counting. To evaluate the magnitude of this problem we emulate the changes in foci counting if the fluorescent points are not convolved with the microscope PSF. This is analogous to what would be seen if you were able to perform a perfect deconvolution on the microscope image, essentially mitigating the microscope blurring effect. The impact of deconvolution can be seen in Fig. 6, where we have compared the foci detected for the actual system, the DSB and γ-H2AX marker visualisations and the DSB and γ-H2AX marker with

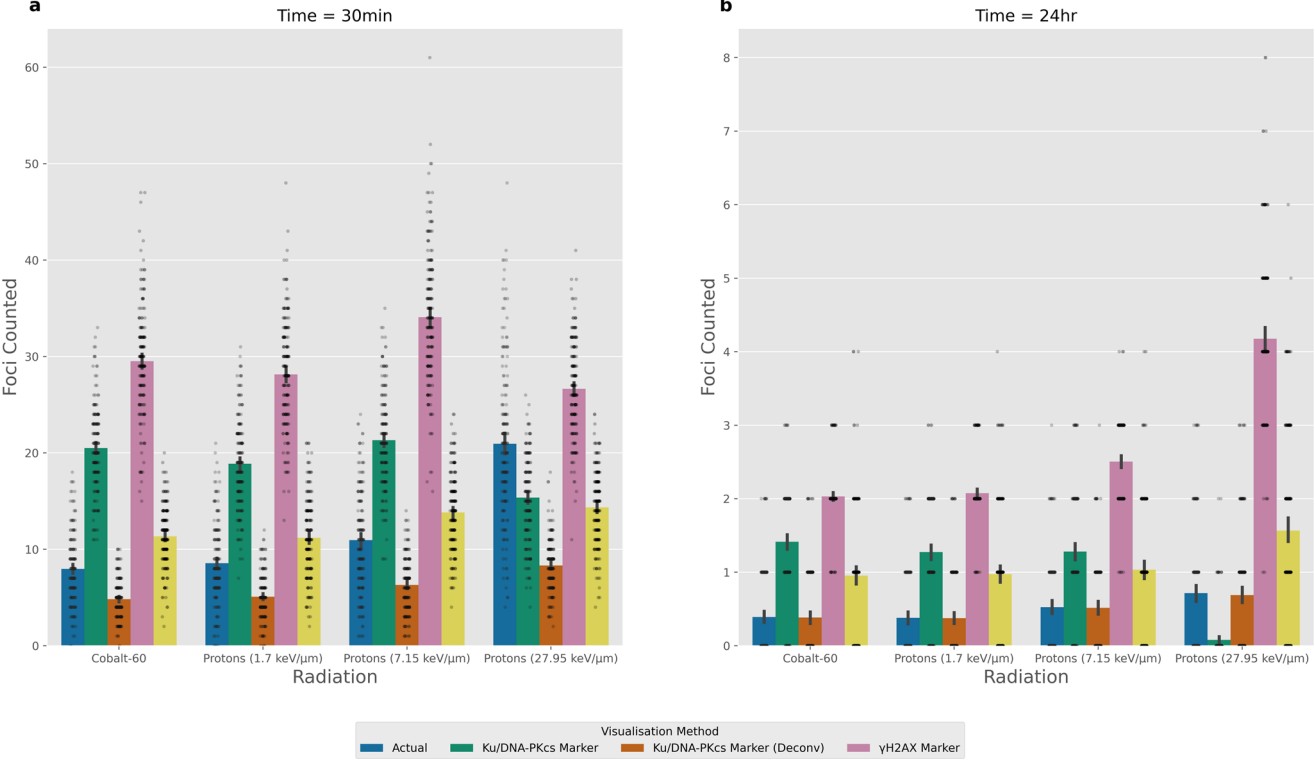

**Fig. 6 Comparison of visualisation methods including deconvolution.** This compares the number of foci detected for the four radiation types delivering 1 Gy at two-time points. Where **a** and **b** is 30 min and 24 h post-irradiation respectively. The actual value is given by the blue bar which is the known simulated number of DSBs within the evaluated z-slice. Whereas, the other bars represent the foci detected from the corresponding visualisation method. Marker visualisation which has been deconvolved represent perfect deconvolution as the image is created before the microscope PSF is applied. Each bar is the averaged result of 200 independent samples and the error bars are the 95% confidence intervals. All data are for an emulated Airyscan ×63 microscope.

deconvolution visualisations. This was compared for the Airyscan ×63 microscope at the 30 min (Fig. 6a) and 24 h (Fig. 6b) time points for 1 Gy of dose. We see that at the 30 min time point of the high-LET protons is usually under-counted in all but the γ-H2AX marker. For the other radiation types tested using a deconvolved DSB or γ-H2AX marker gave better agreement with the actual number of DSBs. This remains the case at 24-h with the deconvolved versions of visualisation giving better agreement than their counterparts which include the convolved microscope PSF. Furthermore, we see that at 24-h the overall best agreement is seen when using a Ku/DNA-PKcs marker that has been perfectly deconvolved for all radiation types. This is due to the Ku/DNA-PKcs marker being a direct indicator of DSBs and the reduced number of DSBs at 24 h reduces the likelihood of breaks clustering together into a single foci. These results help confirm that the fluorescence from neighbouring slices is a major driver of miscounting in confocal imaging.

**Impact of 3D foci analysis on miscounting.** A different approach to reducing miscounting due to influence from neighbouring slices is to analyse the whole 3D stack of the cell. When comparing the single slice analysis to the 3D stack analysis we observe reduction of neighbouring slice influence on over-counting, but the under-counting from spatial resolution limitations remains. This is shown clearly in Fig. 7a, where all 3D analyses showed under-counting when compared to the actual number of DNA DSBs present in the simulation. Furthermore, as LET increases in the proton simulations, we observe the severity of the under-counting increases, which aligns with this effect being caused by spatial resolution limitations. The effects of

under-counting are increased when visualising γ-H2AX markers when compared to Ku/DNA-PKcs markers. At the 24 h time points and lower LETs (Fig. 7b) the Ku/DNA-PKcs marker 3D foci counting does a good job at matching the actual number of DNA DSBs. However, there still persists some under-counting for higher LET radiations. This identifies that whilst analysing the 3D stack was successful in reducing the influence from neighbouring slices the uncertainties related to the underlying spatial resolutions are still present.

**Underlying drivers of miscounting.** It has been identified that both marker type and visualisation can play a major role in the amount of possible miscounting and the subsequent repair kinetics you can arrive at given a particular radiation set-up. Most of the differences between radiation types are being driven by the spatial distribution of the DSBs. Therefore, we calculate the clustering of the DSBs at all the time points, doses and radiation types. This clustering metric allows us to unify over all of these different parameters and compare the systems based on the DSB spatial distribution alone. The clustering is a measure of the average number of DSBs within proximity to each DSB of the system. This proximity has been defined as 200 nm in this study, but similar trends are seen at both 100 nm (Fig. S11) and 500 nm (Fig. S12). We then compare the amount of miscounting for different clustering values and look for an overall pattern (Fig. 8). We see that across all visualisation approaches an increased clustering value equates to increasing levels of under-counting. These effects are reduced in the γ-H2AX marker (Fig. 8b) where the under-counting due to clustering appears to balance with the increased contribution of fluorescence from neighbouring slices

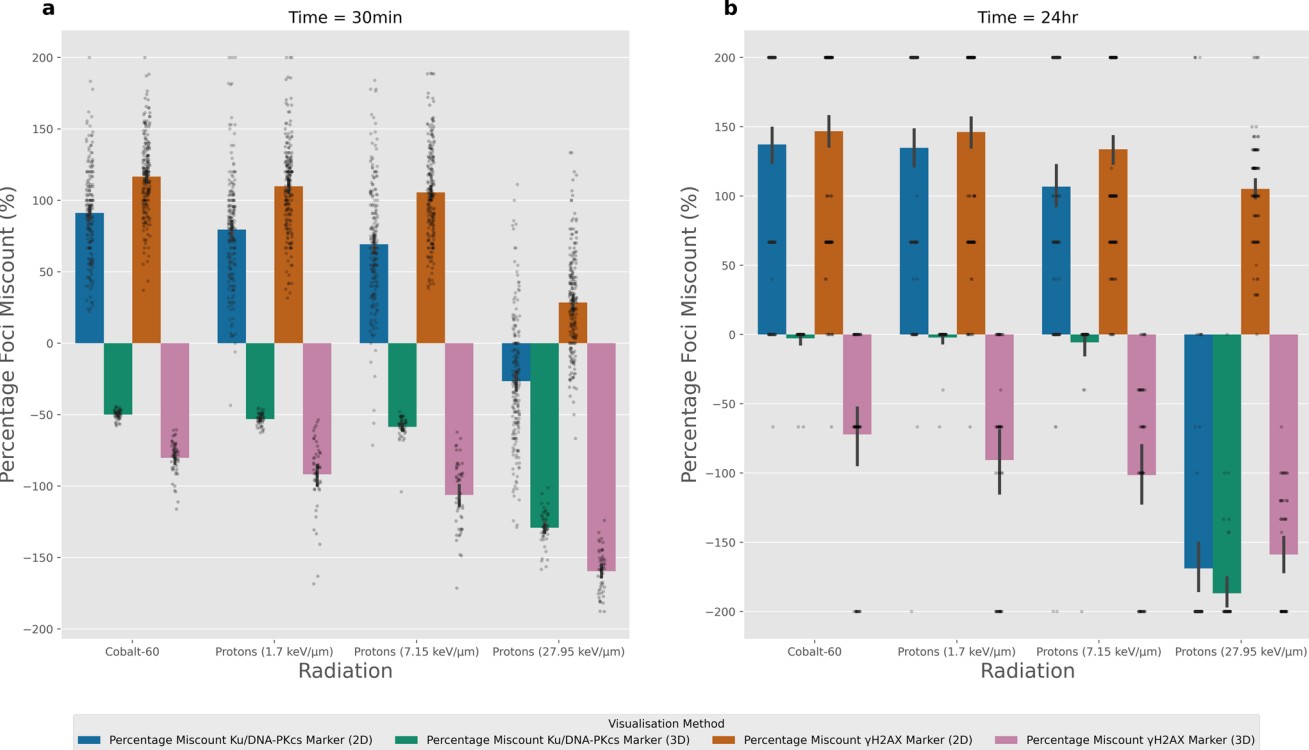

**Fig. 7 Percentage foci miscount when using a single slice and 3D stack microscopy.** This compares the number of foci detected for the four radiation types delivering 1 Gy at two-time points. Where **a** and **b** is 30 min and 24 h post-irradiation respectively. The percentage difference is calculated as the (counted − simulated/((counted + simulated)× 0.5)× 100. 3D foci counting consistently under-counts foci at the 30 min time point with its magnitude increasing as a function of LET. Whereas, 3D foci counting appears to correctly count the small number of foci left at 24 h with an exception of consistent under-counting at high LET. Each bar is the averaged result of 50 independent samples and the error bars are the 95% confidence intervals. All data are for an emulated Airyscan ×63 microscope.

being identified as foci. This can be confirmed by the corresponding deconvolved *γ*-H2AX marker system (Fig. 8d) where the contribution from neighbouring slices are reduced and the pattern of increased under-counting at higher clustering values becomes apparent again. The effects in the Ku/DNA-PKcs marker is similar, but with a smaller contribution of neighbouring fluorescence being counted as foci (Fig. 8a), but a still noticeable increase in under-counting in the corresponding deconvolved system (Fig. 8c).

## Discussion
In this study, the impact on foci miscounting has been evaluated for a range of radiations, doses, microscopes and magnifications. Through the modelling of both the DNA DSB positioning and the emulation of viewing through a microscope, we have been able to leverage a unique position of being able to compare the detected foci to the simulated ground truth. This provides insight into the ability to resolve DNA DSBs when using a fluorescence marker that is subjected to a microscope point spread function (PSF). It has been shown that even under ideal simulated experimental conditions there are significant differences in the amount of miscounting you should expect between different radiations (Co-60 and protons at varying LET), shown in Figs. 1 and 2. This can result in the perceived change of repair kinetics, even when these are simulated as constants (Fig. 3). This brings into question how best to evaluate changes between test groups of experimental work, given that these differences in miscounting are usually omitted. This work has highlighted that the main sources of foci miscount arrive from either over-counting DSBs from neighbouring slices or under-counting DSBs which cannot be distinguished from one another. Whilst counting breaks from other

slices is not inherently unwanted as this damage does exist within the cell, the rate at which this happens varies based on the radiation quality leading to a non-systematic experimental set-up uncertainty. For example, while it is widely thought that high-LET protons may cause additional complex DNA damage which requires increased time for DNA repair[6,7,32] we show the increased under-counting in *γ*-H2AX simulated images can lead to the visual perception of slower repair kinetics caused by miscounting alone. It becomes important to think of how we might disentangle changes due to the uncertainty of the technique and the underlying biology.

This work has predominantly focused on confocal microscopy and viewing a single z-slice. We have included some 3D foci analysis due to the increased amount of multi-slice imaging and 3D foci detection[33–35], especially when using super-resolution microscopy. However, radiation based foci counting is still done using single slice analysis. The developed PyFoci software is well positioned to benchmark the effects of foci miscounting in both 2D and 3D imaging sets. The variation in foci detection technique is increasingly difficult to replicate due to the diverse available methodologies. Therefore, this work attempted to perform a widely available approach with the Laplacian of Gaussian (LoG) technique prioritising consistency over optimisation. This consistency is at the backbone of the evaluations we have made, but we acknowledge that counting accuracy may be improved if the foci counting technique was optimised specifically for an individual setup (microscope, magnification, radiation type and dose). Furthermore, it is also possible to incorporate some element of labelling efficiency into the simulations, in the presented study this was assumed to be 100%, but is likely to vary across labs and equipment. To enable this work to be more representative of an

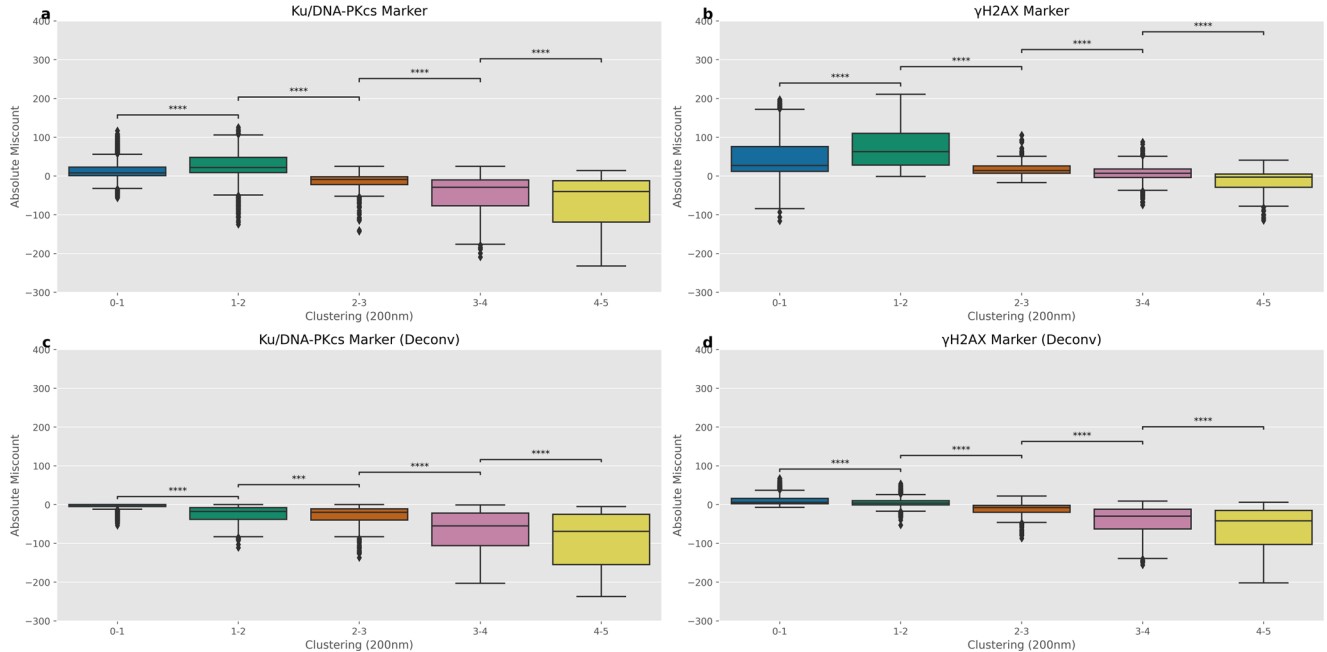

**Fig. 8 Categorising the magnitude of absolute miscounting as a function of DSB clustering.** The values of clustering are calculated based on the known amount of DSBs within the cell nucleus at a given time point. Clustering equates to the average number of DSBs within proximity to each DSB within the simulated cell nucleus. Where proximity for this study is characterised as 200 nm. Clustering is increased for early time points, higher dose, higher LET and is decreased for the opposite. Therefore, the clustering combines time, dose and radiation type parameters into a single metric. **a, b** are for the DSB and $\gamma$-H2AX markers respectively. **c, d** are for the DSB and $\gamma$-H2AX markers with perfect deconvolution respectively. Error bars correspond to 1.5 times the interquartile range in either direction. Mann–Whitney test between categorised clustering values to highlight statistically significant differences. $p$ values have been adjusted using the Bonferroni correction. Significance notation refers to the following thresholds: ns $= p > 0.05$, $*p > 0.01$, $**p > 1e-3$, $***p > 1e-4$, $****p < 1e-4$. All microscope images have been emulated using the Airyscan ×63 point spread function.

individual's experimental approach we have provided the DSB and $\gamma$-H2AX marker distributions which can be visualised using the PyFoci software. This allows the same images (including 3D stacks) to be evaluated using foci detection system better matched to one's experiments.

In modelling the phosphorylation of H2AX some simplifications were made. This includes a spatially linear spread of the activation region from a DSB, whereas it is believed that phosphorylation is spread through physical contact rather than propagation along the chromosome[36]. However, without sub-TAD resolution geometry, it is not possible to incorporate these contact points at present. The amount of H2AX is known to be cell-type dependent[4], in this study we use a constant of 10% of H2A histones being the variant H2AX. If this work was to be expanded to measure changes due to cell-type we would recommend that the H2AX variant percentage is altered to best capture any cell-type specific effects[37]. Whilst not included in this study, it has been suggested that DSBs inflicted at the TAD boundary may result in a multi-TAD activation[36], as these regions make up a small proportion of the genome, making the chances of damage induction at these regions small, this effect has not been included. Furthermore, the model currently assumes a stationary break end, where in fact the break ends are known to be mobile, this may potentially increase the congregation of break ends increasing the difficulty to distinctly identify multiple DSBs at a single site. This is confounded by other drivers of motions, such as cell cycle and chromatin condensation which require further geometric modelling developments before they can be accounted for. Finally, although $\gamma$-H2AX marker has been explicitly detailed here with the H2AX histones used as points of fluorescence, it has also been identified[30] that 53BP1 and MDC1 similarly spread across the TAD. Therefore, results for the $\gamma$-H2AX marker may be a good approximation for these markers but would benefit from further

evaluation of how these markers differ to $\gamma$-H2AX. A limitation of this study is that the PyFoci images have not been directly compared with their experimental foci images from the corresponding microscopes being emulated. The next steps of the PyFoci project should be a more direct comparison to experimental and simulated images to ensure what is being emulated is sufficiently similar.

The PyFoci software can be readily applied to any in silico model of DNA damage and repair which can output DNA DSB positions as a function of time. Through the modelling of the microscopy and the inherent uncertainties which come from the technique, it becomes apparent that modellers may prefer to generate these computational microscope images and perform foci counting when trying to compare their models to the experimental data. It is common for modellers to evaluate their DNA repair systems which are representative of the absolute system to experimental data[22,38,39], but this work has identified that the simulated system should be distorted in the same manner as the experimental fluorescence imaging is a distortion on the actual biology. This would allow for better matching between the simulated DNA repair from computational foci imaging to the corresponding experimental foci imaging.

## Methods
**Modelling DNA double-strand break damage distributions and DNA repair kinetics.** To generate a model of the distribution of the DNA damage radiation was simulated through a polymer bead model of the organised genome. The genome organisation is derived from modelling the TAD contact probabilities of Hi-C data as a 3D polymer with each bead representing a single TAD. To get a spread of representative genome structures the Monte Carlo Markov-Chain model G-NOME[29] was used to produce 200 different IMR90 (normal lung fibroblast) geometries from published Hi-C data by Rao et al.[40], (GEO Accession GSE63525). Each geometry was solved as a flattened ellipsoid (11.8 × 11.8 × 1.0 µm radii) to be representative of cells that have been plated for microscopy[41].

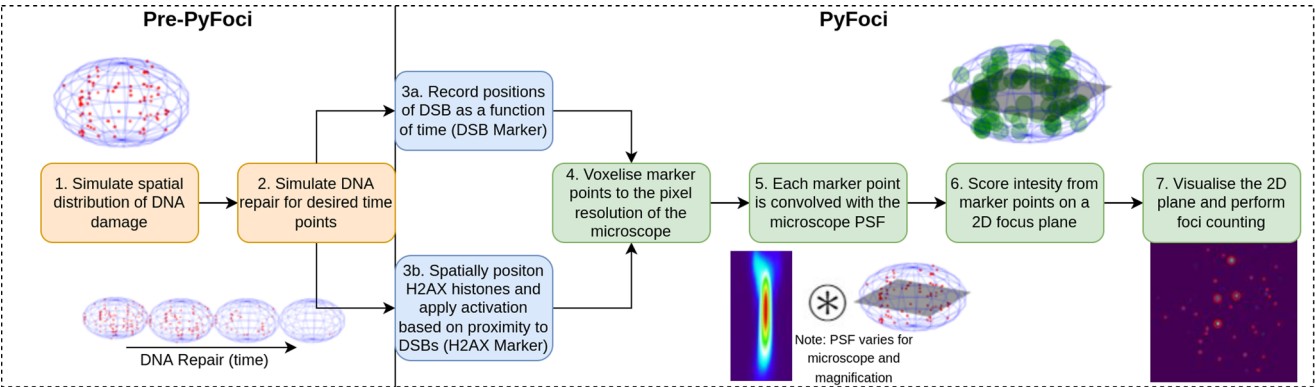

**Fig. 9 Schematic overview of creating computational microscope images using PyFoci.** The generation of simulated DNA DSBs and repair (steps 1–2) is independent of PyFoci and can be interchanged with other models. Any model which can output DSB position as a function of time can use PyFoci to create computational microscope fluorescent foci images (for at least the Ku/DNA-PKcs marker visualisation). Steps 3a and 3b represent the choice in the visualisation approach, whilst Ku/DNA-PKcs marker only requires information on the DSB spatial distribution, the γ-H2AX marker requires information on the chromosome geometry, namely the topologically associated domains the DSBs are created in. Steps 4–7 are the same regardless of the visualisation marker, the only difference is what biological object is being fluoresced (DSBs or γ-H2AX). Step 5 requires the microscopes PSF to be defined for the desired microscope and magnification under evaluation. Step 6 requires the user to define at what time point they wish to create the microscope image and at what z-axis slice. Step 7 creates the image and performs the foci counting on the produced image, this can be done with the in-built foci counter or can be exported to external foci counting software.

To model proton irradiation, the track structure of the incident protons was simulated using the Monte Carlo toolkit Geant4 (v10.5.1)[42] with the default Geant4-DNA physics list[23]. Each track is simulated as a series of interaction limited steps, with each step accounting for the changes in energy and trajectory of both the primary and secondary particles, resulting in spatial information of energy deposition from the incident protons. To convert energy depositions into DNA strand breaks several conditions must be met. Firstly, the energy depositions must have occurred in the genome represented by the polymer beads. Secondly, a spatial sampling of 14.1% is applied to the bead to account for DNA sparsity within a TAD and reproduces DSB yields seen in our previous work[18]. Finally, an energy-based probability of break induction (0 at 5 eV to 1 at 37.5 eV) is applied for energy deposited in a backbone molecule, which is similar to other works[43] and incorporates the possibility of DNA damage occurring below the ionisation threshold of DNA[44,45]. If all conditions are passed the energy deposition is accepted as a strand break and is randomly assigned to a strand of the double helix with equal probabilities. The equivalent position along the chromosome for the damage is randomly selected from the base pair range of the TAD the damage occurs in. Once all strand breaks are calculated from the simulated incident proton irradiation a clustering algorithm is used to distinguish which strand breaks are likely to form DSBs. The classification of DSB is given to strand breaks that are on the opposite strand and separated by 3.2 nm or less (equivalent to 10 bp).

To model photon irradiation, DSB induction is assumed to follow a Poisson distribution with an average of 30 DSBs/Gy. The damage is modelled through the same 200 Hi-C solved genome models as proton irradiation. Each DSB is assigned a chromosome at random, weighted by the chromosome size relative to the total genome size. The same process is applied to assigning a bead, which is weighted according to bead size. A random X, Y, Z position within the selected bead is also applied. All DNA damages are recorded in the standard DNA damage format[46].

After DNA DSB induction we utilised a bi-exponential repair kinetics model which was applied across all radiation simulations to create a consistent repair rate (Fig. 9). This was uniformly applied so that only the difference caused by miscounting would be observable in the resultant repair kinetics. The bi-exponential repair kinetics was a normalisation of the γ ray fit from ref. [32], so it could be applied to any number of initial induced DSBs (Eq. (1)):

$$\lceil N(t) \rceil = N_0 (a_1 e^{-t/\tau_1} + a_2 e^{-t/\tau_2}) \tag{1}$$

where $\lceil N(t) \rceil$ is the ceiling integer of the number of DSBs at a given time $t$ hours and $N_0$ the initial number of DSBs. The four constants of fit are $a_1 = 0.711$, $a_2 = 0.289$, $\tau_1 = 1.54$ h, $\tau_2 = 10$ h.

**Generating PyFoci microscope images.** A schematic overview of the whole process to create the computational microscope images are shown in Fig. 9. The spatial distribution of the damage as a function of time after irradiation is derived from the bi-exponential repair model. These files can be read by PyFoci which maps the DNA DSBs to the same resolution (pixel spacing) of the microscope being used. The microscope PSF is derived experimentally through the measurement of sub-resolution fluorescent beads of a known size, allowing for the PSF to be computed (distilled). The DSB positions are used as a point source, due to the typical size of antibody molecules (~10 nm)[31] being smaller than the resolution of the microscope. The DSBs are then convolved with the microscope's PSF to

emulate the intrinsic blurring of a light source from microscopy in 3D. This includes the amount of out-of-plane fluorescence which is not blocked by the confocal pinhole. The light is then scored in either a single 2D matrix at a set z-value to emulate a z-slice or a 3D matrix over a range of z-values to emulate a z-stack from a confocal microscope. The resultant matrix can then be viewed as a computational microscope image. These steps are more representative of using a DSB-marker which directly attaches to the repair proteins (e.g. Ku70/80 or DNA-PKcs) found at break ends[47], rather than indirect markers such as γ-H2AX or 53BP1. The direct markers will be referred to as Ku/DNA-PKcs Markers and the indirect markers referred to as γ-H2AX Markers throughout.

To evaluate indirect markers a histone based model was deployed. The number of H2AX histones required for placement was based on assuming 10% of H2A histones being the H2AX variant[48], each nucleosome having two copies of the H2A histone and each nucleosome being 146 basepairs[49]. The histones were randomly placed within the polymer beads (which represent TADs) which had DSBs present. Utilising the findings of Arnould et al.[30], it was assumed that activation of H2AX was restricted within the TAD the DSB belonged to (Fig. S1). The strength of the H2AX activation was also taken from Arnould et al.[30], and was fitted using a Cauchy-Lorentz distribution (Eq. (2)) based on the Chip-Seq measurements of γ-H2AX read counts as a function of distance in Mbp from DSB (Fig. S2):

$$\text{Activation} = 0.23 + \frac{0.38}{1 + 4\left(\frac{\text{Distance}}{0.45}\right)^2} \tag{2}$$

As the sub-TAD organisation is not present within the model, the effective base pair distance was calculated by the Euclidean distance between the break and histone, where the conversion was based on the diameter of the TAD bead divided by the genomic content of the same bead. Each histone was given a value of activation based on its proximity to the DSB which would represent the intensity of fluorescence. Nearby breaks which share histones to activate were assumed to activate the histone in an additive manner. This was carried out for all DSBs and was visualised in the same manner as the DSB-marker method within PyFoci. This allows the production of γ-H2AX-marker based images to compare to the DSB-marker images.

The microscopy simulation of both marker types assumes the perfect efficiency of the fluorescent antibody to attach to its molecular target. In total six microscopes were used each with several magnifications. The microscopes utilised were: Carl Zeiss LSM880 with Airyscan fast mode (Airyscan), Leica sp8 TCS inverted confocal with STED super-resolution mode (gSTED), Carl Zeiss Axiovert 200 M (lowlight), Leica sp8 TCS upright confocal (MultiPhoton), Perkin Elmer Opera Phenix High Content Screening System (Phenix) and super-resolution mode on gSTED system (STED). A PSF was measured for each microscope magnification combination resulting in a total of 24 variations of microscope blurring. The PSFs are distilled and provided in the H5 hierarchical data format (*.h5) files which can be directly parsed into PyFoci. The pixel spacing and NA for each microscope combination are given in Table 1, with the full width half maximum of the central axis for each PSF is given in Table S1. Example image set demonstrating how the different marker type, microscope, magnification and time impacts the visualised image has been provided within the available datasets.

**Automated foci counting on generated microscope images**. Computationally generated microscope images were generated for each of the 200 solved nuclear geometries at 16 radiation set-ups, 23 microscope magnification configurations and six-time points (0 s, 15 min, 30 min, 2 h, 6 h, 24 h) as a single z-slice centrally positioned in the cell nucleus. This resulted in a total of 441,600 computationally generated microscope images for both the DSB-marker and γ-H2AX-marker visualisation. Therefore, an automated foci counting approach was used to count the number of foci in all 883,200 images and compare the number of identified foci to the number of known breaks within the same slice. The automated foci detection was carried out using a LoG technique found in the Python package scikit-learn (v0.23). The LoG technique relies on the user set minimum, maximum and number of sigma, along with a threshold value. The various sigma parameters control the allowed size of detected foci and the threshold value determines the lower bound of the intensity required for detection. To be consistent in the foci detection the parameters were fixed for foci detection across all simulation set-ups within the same visualisation method (DSB-marker and γ-H2AX-Marker). Due to the intrinsic differences between the fluorescent component of the DSB-marker and γ-H2AX-marker images, there were two distinct sets of foci counting parameters for the visualisation methods. As the intensity values change drastically for different radiation parameters, the threshold was set as a percentage of the maximum intensity value of the computational microscope image. This maximum intensity value was calculated for each simulation set-up at the 15 min time point and was set for all other time points, this was to ensure that the automated detection did not have a chance of increased foci detection at later time points. The parameters were optimised by matching the number of detected foci a subset of images from all the set-ups with a user manually counting foci. The LoG parameters used for the DSB-marker are as follows: $minsigma = 2$, $maxsigma = 20$, $numsigma = 1$, $threshold = 8\%$. The LoG parameters used for the γ-H2AX-marker are as follows: $minsigma = 2$, $maxsigma = 20$, $numsigma = 3$, $threshold = 4\%$. These LoG parameters are constant between both 2D and 3D foci counting analyses. Example image set of automatic foci detected images has been provided within the available datasets.

**Statistics and reproducibility**. To test if the difference in miscounting was significant between different doses and radiation types a Mann–Whitney test was used. The reported $p$ values were adjusted using the Bonferroni correction to reduce type I error. These calculations were performed using the statistical tests in python's statannot package (v0.2.3) which utilise the SciPy package (v1.6.2) stats methods. Within the text $p$ value thresholds have been categorised as ns = $p > 0.05$, *$p > 0.01$, **$p > 1e-3$, ***$p > 1e-4$, ****$p < 1e-4$. All sample sizes along with the corresponding microscope and magnification emulated is given in the figure captions. Explicit $p$ values for every statistical test carried out can be found in the figshare repository in the "Explicit_PValues.zip" (see Data Availability statement for link). All plotted data can be found pre-processed (i.e. foci counted) in the "FociCountedDatasets.zip" and "FociCountedDatasets_3DAndDeconv. zip" in the figshare repository. All raw data, including the data to make microscope images and the microscopes images created in this work, is provided in the figshare repository.

**Reporting summary**. Further information on research design is available in the Nature Research Reporting Summary linked to this article.

## Data availability

The datasets generated during and/or analysed during the current study are available in the figshare repository, https://doi.org/10.48420/14398790.

## Code availability

The PyFoci source code is available at https://gitlab.com/PRECISE-RT/releases/pyfoci (https://doi.org/10.5281/zenodo.6513747). A demonstration Google Colab notebook has been created for a user-friendly experience to visualise and analyse all the data in this study, available at https://github.com/SamPIngram/PyFoci_Colab.

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

## Acknowledgements

The authors would like to acknowledge the help of Kang Zeng and Steven Bagley from the Cancer Research UK Manchester Institute for obtaining the microscope point spread functions and their insight into microscopy. The authors would like to acknowledge the assistance given by Research IT and the use of the Computational Shared Facility at The University of Manchester. This work was supported by the NIHR Manchester Biomedical Research Centre (BRC-121520007 to K.J.K., R.I.M., N.F.K. and M.J.M.); the STFC Global Challenge Network+ in Advanced Radiotherapy (ST/N002423/1 to S.P.I., M.J.M., K.J.K. and R.I.M.); the European Union Horizon 2020 Research and Innovation (730983—INSPIRE to S.P.I., N.F.K., K.J.K., R.I.M., M.J.M., N.T.H. and J.W.W.); the United Kingdom Engineering and Physical Science Research Council (EP/S024344/1—BioProton to S.P.I., K.J.K., R.I.M., N.F.K. and M.J.M.); a UK Research and Innovation Future Leaders Fellowship (MR/T021721/1 to S.J.M.); the National Institute of Health (NIH)/National Cancer Institute (NCI) (R01 CA187003—TOPAS-nBio to J.S.); the funders had no role in study design, data collection and analysis, the decision to publish, or preparation of the manuscript.

## Author contributions

S.P.I., J.W.W., N.T.H., M.J.M., S.J.M. and J.S. designed the project. S.P.I. created the code, generated the results and wrote the manuscript. S.P.I. and N.T.H. generated the DNA damage data. A.L.C. and E.E.S. critiqued the manuscript based on experimental radiobiology. N.F.K., R.I.M., K.J.K. and M.J.M. provided day to day supervision of the work. All authors revised and agreed upon the content of the manuscript.

## Competing interests

The authors declare no competing interests.
