## [Peer Review File · Communications Biology]

Reviewers' comments:

Reviewer #1 (Remarks to the Author):

The manuscript by Ingram and colleagues proposes a computational approach for the quantification of miscounting in radiation induced DSB as measured by means of immunofluorescent foci. According to my experience in the field, this is a timely contribution. Despite the use of immunofluorescent foci became very popular in radiobiology in the last decades, I believe that this interesting tool is often prone to misuse, due to a number of reasons that confuse the link between actual DSB and measured foci, that essentially represent the motivation of this work. From this point of view, I appreciated the efforts performed by the Authors, and I am confident that their results might be of interest for a comparably large community. Concerning more technical aspects, the approach adopted is sound, and the statistical analysis seems to be consistent. The language of the paper is good, despite some misspelling distributed over the text.

I have few specific comments, which are addressed below:

- I found confusing and not 100% accurate referring to Ku/DNA-PKs foci as "DSB marker". Indeed, gH2AX is also essentially a DSB marker, even though less direct compared to Ku/DNA-PKs. I would wonder what is the experimental efficiency of Ku/DNA-PKs in identifying induced DSB. Moreover, this gets clear only at the end of the paper, where the Methods Section is presented. I would recommend the Authors referring to "Ku/DNA-PKs" and to "gH2AX" foci, avoiding the confusing use of DSB marker. A short explanation of the difference between the two markers could appear somewhere else in the text.
- The PyFoci software presented by the Authors is a free software and this could turn out to be useful for several users. From this point of view, the Authors should comment more on the calibration process of this software, and on the most critical parameters to be set-up. I imagine that a proper calibration is a crucial step towards a possible generalised use of this software or, more in general, of the presented approach.
- Please, enlarge figure legends and symbols, which are hard to read in the present form.

Reviewer #2 (Remarks to the Author):

The manuscript by Ingram et al. establishes a novel computational approach to help elucidate impact of miscounting of radiation-induced DSB foci. This study provides a unique software described here as PyFoci that is able to emulate the fluorescent foci labelling both direct DNA DSB or indirect gamma-H2AX foci. This novel tool is built upon previously validated models. However, unlike previous models, the tool provides an unprecedented platform for quantifying miscounting between the actual number of DSBs present within the simulation and number of DSBs foci counted on the emulated microscope image. The approach is fundamentally a game changer for the DNA repair research field and for Radiation Biodosimetry. PyFoci may help to elucidate the deviation between absolute DSBs and the number of DSBs foci obtained in a fluorescence-based imaging. To the physiological standpoint, this approach may help gain an in-depth assessment of actual DNA repair kinetic and repair process.

While the study is novel and comprehensively described for the radiation biology/DNA repair audience, it is certainly a work most suited for a specialty journal.

The authors should address the points below:

1. Microscopic images for most of the quantifications presented in this study (figure 1 to 7) should be provided to allow readers for better assessing accuracy of the data presented. The lack of representative images makes it harder to offer an independent assessment.
2. As shown in most figures, but particularly in figure 3, the largest miscount for the DSB marker and gamma-H2AX marker was seen for high-LET proton. In fact, the authors claim that for most instances, the fluorescence from neighboring slices are the major driver of miscounting in confocal imaging. The argument seems to underestimate the idea that high-LET protons may cause additional complex DNA lesions beyond the typical DSB (detected by DSB marker here), while these additional lesions may also be detected by gamma-H2AX, and therefore may be the reason of the discrepancy between the actual DSB marker and gamma-H2AX foci. The use of gamma-H2AX as a single marker for the entire study is certainly not sufficient. The authors should include

co-staining with 53BP1 (MDC1 might not be a good option given its pattern of foci in microscopy). Also, to ascertain for contribution from neighboring slices, how would the miscounts pattern look like if the authors choose to perform some of the experiments in condition where there is a minimal influence from neighboring slices (ie minimizing cell number?).

Reviewer #3 (Remarks to the Author):

The authors present a simulation tool, which can be used to estimate the under or over counting of foci in a 2D imaging approach.

The method used seems to be valid, but the use of 2D images is well known to be non-accurate and therefore it should not be the targeted experimental method. Indeed the 3D counting of foci is state of the art, as there all information is gathered and used for counting. Therefore the simulation method should be applied, at least also for 3D images.

Furthermore, no real microscopic images are shown. This is urgent for the reader to judge whether the method is working well. A comparison between experimental images and simulated images should be given.

The simulation is considered to give the truth, which is not the case. Simulation tools can only be as good as the data used for developing the programm. Therefore a variation of underlying simulation parameters should be done and the effect to the results should be shown.

Additionally the manuscript would profit from spelling and grammar proof reading.

The supplementary figures 1 and 2 are not properly referenced in the manuscript.

We thank all reviewers for their feedback. We have carefully reviewed each comment and have tried to address each fully. Please see the responses below.

Reviewer comments will be in this colour and italicised

Author's response will be in this colour and bold

Reviewer #1 (Remarks to the Author):

The manuscript by Ingram and colleagues proposes a computational approach for the quantification of miscounting in radiation induced DSB as measured by means of immunofluorescent foci. According to my experience in the field, this is a timely contribution. Despite the use of immunofluorescent foci became very popular in radiobiology in the last decades, I believe that this interesting tool is often prone to misuse, due to a number of reasons that confuse the link between actual DSB and measured foci, that essentially represent the motivation of this work. From this point of view, I appreciated the efforts performed by the Authors, and I am confident that their results might be of interest for a comparably large community.

Concerning more technical aspects, the approach adopted is sound, and the statistical analysis seems to be consistent. The language of the paper is good, despite some misspelling distributed over the text.

I have few specific comments, which are addressed below:

- I found confusing and not 100% accurate referring to Ku/DNA-PKs foci as "DSB marker". Indeed, gH2AX is also essentially a DSB marker, even though less direct compared to Ku/DNA-PKs. I would wonder what is the experimental efficiency of Ku/DNA-PKs in identifying induced DSB. Moreover, this gets clear only at the end of the paper, where the Methods Section is presented. I would recommend the Authors referring to "Ku/DNA-PKs" and to "gH2AX" foci, avoiding the confusing use of DSB marker. A short explanation of the difference between the two markers could appear somewhere else in the text.

We have now amended the text to use the nomenclature above to go along with the description of both markers in the "Generating PyFoci Microscope Images" methods section.

- The PyFoci software presented by the Authors is a free software and this could turn out to be useful for several users. From this point of view, the Authors should comment more on the calibration process of this software, and on the most critical parameters to be set-up. I imagine that a proper calibration is a crucial step towards a possible generalised use of this software or, more in general, of the presented approach.

Thank you for this comment. We have attempted to help navigate users through using the software by creating an interactive notebook using Google Colab. This will allow users to see the calibration process and adjust it to their needs for easier benchmarking. The link to the notebook is here: https://colab.research.google.com/github/SamPIngram/PyFoci_Colab/blob/main/PyFoci.ipynb and is also shown in the code availability section.

- Please, enlarge figure legends and symbols, which are hard to read in the present form.

We have enlarged figure legends where possible and have increased the DPI of some of the harder to see figures.

Reviewer #2 (Remarks to the Author):

The manuscript by Ingram et al. establishes a novel computational approach to help elucidate impact of miscounting of radiation-induced DSB foci. This study provides a unique software described here as PyFoci that is able to emulate the fluorescent foci labelling both direct DNA DSBs or indirect gamma-H2AX foci. This novel tool is built upon previously validated models. However, unlike previous models, the tool provides an unprecedented platform for quantifying miscounting between the actual number of DSBs present within the simulation and number of DSBs foci counted on the emulated microscope image. The approach is fundamentally a game changer for the DNA repair research field and for Radiation Biodosimetry. PyFoci may help to elucidate the deviation between absolute DSBs and the number of DSBs foci obtained in a fluorescence-based imaging. To the physiological standpoint, this approach may help gain an in-depth assessment of actual DNA repair kinetic and repair process.

While the study is novel and comprehensively described for the radiation biology/DNA repair audience, it is certainly a work most suited for a specialty journal.

The authors should address the points below:

1. Microscopic images for most of the quantifications presented in this study (figure 1 to 7) should be provided to allow readers for better assessing accuracy of the data presented. The lack of representative images makes it harder to offer an independent assessment.

We thank the reviewer for this point. As it is impractical to display all 800k+ images used in the analysis of some of the figures. We have added a subset of example images in the data store hosted on Figshare (https://figshare.com/ndownloader/files/30669551?private_link=ae99c2d5888604e16eec). Furthermore, we have now included an interactive notebook through Google Colab which allows the user to generate every image in this study and even allows them to perform the foci counting with the study parameters or their own to allow for this independent assessment. Finally, the notebook also allows for image export where it can be imported into the user's own foci counting software for evaluation of miscounting more specific to their own lab procedures. The link to the notebook is here: https://colab.research.google.com/github/SamPIngram/PyFoci_Colab/blob/main/PyFoci.ipynb and is also shown in the code availability section.

2. As shown in most figures, but particularly in figure 3, the largest miscount for the DSB marker and gamma-H2AX marker was seen for high-LET proton. In fact, the authors claim that for most instances, the fluorescence from neighboring slices are the major driver of miscounting in confocal imaging. The argument seems to underestimate the idea that high-LET protons may cause additional complex DNA lesions beyond the typical DSB (detected by DSB marker here), while these additional lesions may also be detected by gamma-H2AX, and therefore may be the reason of the discrepancy between the actual DSB marker and gamma-H2AX foci. The use of gamma-H2AX as a single marker for the entire study is certainly not sufficient. The authors should include co-staining with 53BP1 (MDC1 might not be a good option given its pattern of foci in microscopy). Also, to ascertain for contribution from neighboring slices, how would the miscounts pattern look like if the authors choose to perform some of the experiments in condition where there is a minimal influence from neighboring slices (ie minimizing cell

number?).

We thank the reviewer for this key point. The sections where we examine miscounting under perfect deconvolution (i.e., stopping the fluorescent contribution from other slices) is how we demonstrate this influence. Furthermore, we believe the new addition of the 3D analysis also helps highlight these impacts and the benefit 3D foci counting can have on minimising this issue. It is also important for us to clarify that all results are simulated including the gamma-H2AX marker. The distinction between the two markers, DSB (now called Ku/DNA-PK) and gamma-H2AX markers, is that they are direct and indirect markers of DNA DSB, respectively. As the results are simulated, we can show the levels of miscounting in an idealistic view, where there is 100% labelling efficiency (we have added a sentence in the discussion to point this out), negating the requirement of co-staining. However, we agree that for experimental work co-staining is needed to account for robust identification of DNA DSBs.

Reviewer #3 (Remarks to the Author):

The authors present a simulation tool, which can be used to estimate the under or over counting of foci in a 2D imaging approach.

The method used seems to be valid, but the use of 2D images is well known to be non-accurate and therefore it should not be the targeted experimental method. Indeed the 3D counting of foci is state of the art, as there all information is gathered and used for counting. Therefore the simulation method should be applied, at least also for 3D images.

We thank the reviewer for this point. Whilst it was something we initially wanted to avoid due to the large amount of analysis still being performed on 2D image sets we do agree that 3D foci counting is becoming more abundant. Therefore, we have expanded the study to include a brief comparison of miscounting between 2D and 3D foci counting under the “Impact of 3D Foci Analysis on Miscounting” section of the results. We have also updated the methods and discussion sections to reflect this addition. Furthermore, this has added the native support for PyFoci to do this with any of our datasets which is something that can be leveraged by interesting parties using the now included Google Colab notebook.

Furthermore, no real microscopic images are shown. This is urgent for the reader to judge whether the method is working well. A comparison between experimental images and simulated images should be given.

Unfortunately, direct comparison with real microscopic images is beyond the scope of this project and remains the subject of further work. The study was focused on the development of the toolset and its release to the wider community for benchmarking purposes. We have added this into our discussion as a limitation of the study.

The simulation is considered to give the truth, which is not the case. Simulation tools can only be as good as the data used for developing the program. Therefore a variation of underlying simulation parameters should be done and the effect to the results should be shown.

We agree that it is important to be aware of the limitations of the simulations. The simulations of DNA damage are based on multiple published pieces of work and evaluating the sensitivity of these parameters would be beyond the scope of this work. We do attempt to abstract away some of the simulated parameters when evaluating the effects of miscounting with cluster density which just looks at its sensitivity to spatial clustering of DNA damage rather than an explicit dose or LET. Furthermore, PyFoci is DNA damage model-independent only relying on the DNA damage distribution rather than any explicit model. The parameters used in the emulation of H2AX, and automated foci counting have been made available to users via the Google Colab notebook. Foci detection tends to be quite subjective so this notebook may be useful for users to define their own counting criteria. Beyond this the notebook also allows export of PyFoci images in various formats which could be used in lab-specific foci counting software.

Additionally the manuscript would profit from spelling and grammar proof reading.

We have endeavoured to correct spelling and grammar mistakes where seen.

The supplementary figures 1 and 2 are not properly referenced in the manuscript.

Supplementary figures are now correctly referenced in manuscript.

REVIEWERS' COMMENTS:

Reviewer #1 (Remarks to the Author):

In the revised manuscript, Ingram and colleagues addressed the issues raised with the first revision. I think that the quality of the paper, which was already good, has been further improved. I therefore endorse the publication of this work.

Reviewer #2 (Remarks to the Author):

The authors have comprehensively addressed my comments in this revised manuscript.

I do however have a minor comment: the paper would greatly profit from increasing front size for most figures.

Reviewer #3 (Remarks to the Author):

The authors adequately changed all mentioned point, which clearly enhances the quality of this work.